# Patient Safety and Staff Well-Being: Organizational Culture as a Resource

**DOI:** 10.3390/ijerph19063722

**Published:** 2022-03-21

**Authors:** Luo Lu, Yi-Ming Ko, Hsing-Yu Chen, Jui-Wen Chueh, Po-Ying Chen, Cary L. Cooper

**Affiliations:** 1Department of Business Administration, National Taiwan University, Taipei 106, Taiwan; ming80706@gmail.com; 2Department of Obstetrics and Gynecology, Taipei City Hospital and Musoon Women’s and Children’s Clinic, Taipei 106, Taiwan; hychen2005@gmail.com; 3Medical Quality Management Center, Taipei City Hospital, Taipei 106, Taiwan; a3330@tpech.gov.tw (J.-W.C.); a0926846825@gmail.com (P.-Y.C.); 4Alliance Manchester Business School, University of Manchester, Manchester M15 6PB, UK; cary.cooper@manchester.ac.uk

**Keywords:** patient safety culture, staff burnout, work–life balance, conservation of resources

## Abstract

The present study examines the relationship between patient safety culture and health workers’ well-being. Applying the conservation of resources mechanism, we tested theory-based hypotheses in a large cross-disciplinary sample (N = 3232) from a Taiwanese metropolitan healthcare system. Using the structural equation modeling technique, we found that patient safety culture was negatively related to staff burnout (β = −0.74) and could explain 55% of the total variance. We also found that patient safety culture was positively related to staff work–life balance (β = 0.44) and could explain 19% of the total variance. Furthermore, the above relationships were invariant across groups of diverse staff demography (gender, age, managerial position, and incident reporting) and job characteristics (job role, tenure, and patient contact). Our findings suggest that investing in patient safety culture can be viewed as building an organizational resource, which is beneficial for both improving the care quality and protecting staff well-being. More importantly, the benefits are the same for everyone in the healthcare services.

## 1. Introduction

Contemporary healthcare institutions face a dual challenge: providing state-of-the-art health care and maintaining a healthy and effective workforce to deliver the high-quality service to patients. Thus, ensuring both patient safety and staff well-being are equally important goals for medical institutions. In the past ten years, healthcare organizations worldwide have invested extensively in various interventions aiming at improving patient safety, whereas measures protecting staff well-being have attracted relatively fewer organizational resources. The potential association between safety culture and staff well-being is seriously understudied, though the scanty research seems to suggest a negative association between patient safety culture and staff burnout [1]. To understand this association, we apply the conservation of resources (COR) theory and purport that a positive and strong organizational culture emphasizing patient safety can act as valuable work resources operating at multiple levels [2]. Together, these resources build a positive work environment which provides staff meaning of work and enables them to better deliver professional service with adequate support and collaboration [3], thus reducing the risk of professional burnout [3,4,5,6]. Furthermore, although existing research disproportionally focuses on physicians and nurses [1,5,6], we believe that the value of safety culture benefits all the staff via resource abundance and cross-disciplinary collaboration. In the present study, we use data from a large cross-disciplinary sample of hospital staff in a Taiwanese healthcare system to test the COR theory-based hypotheses that patient safety culture is uniformly related to employees’ well-being (i.e., burnout and work–life balance), irrespective of demographics and work roles.

## 2. Theoretical Framework and Hypotheses Development

### 2.1. Patient Safety Culture and Staff Well-Being

Safety culture refers to the perceptions, beliefs, values, attitudes, and competencies within an organization pertaining to safety and prevention of harm [7,8]. To ensure patient safety and improve quality of patient care, healthcare institutions usually take a high-level approach focusing on improving organizational processes and work conditions to eradicate institutional and managerial factors that contribute to medical errors. Safety improvement efforts such as medication management, quality circles, and cross-disciplinary teamwork have been invested in medical institutions worldwide. However, a recent review suggested that the overlooked individual-level factors may help to further improve patient safety [1]. Staff well-being, negatively manifested in professional burnout and positively manifested in work–life balance, is an important individual factor to be explored in safety culture research [1,9]. In Taiwan, where our present study was conducted, the hospitals run their clinical service under the unique umbrella of “universal health coverage”, a world-standard high-quality and low-cost national health insurance system (NHIS) that is accessible to every citizen. Despite the different governance and conditions for clinical practices, available cross-country evidence seems to suggest that Taiwanese health workers (e.g., physicians, nurses) suffered no more (or less) burnout than those in other countries [10,11].

Burnout is a prevalent strain response to work stress, especially among workers in the service industrials. Burnout has three main facets, namely emotional exhaustion, depersonalization, and diminished sense of accomplishment, with exhaustion at the center of the syndrome [12]. Work–life balance of healthcare workers is believed to root in the organizational climate which encourages staff to tend to their self-care needs such as taking recovery breaks, health diets, and regular exercise [9]. Studies examining the relation between staff burnout and patient safety culture are scant, with only two identified in the latest review of the literature [1,13,14]. Both studies used burnout scores to “predict” safety perception and errors reporting, with cross-sectional data. One rare longitudinal study, however, found an intriguing interdependent relationship between medical errors and strains among internal medicine residents [15]. Specifically, self-perceived medical errors were significantly associated with doctors’ quality of life, burnout, and depression at a subsequent time. High burnout and diminished empathy were related to increased risk of future errors. These findings suggest that risk for patient safety (perceived errors) may be a “cause” for doctors’ burnout. To contribute to the mainstream research on healthcare quality, we adopt the same conceptualization and measurements of staff well-being, burnout, and work–life balance as commonly used in the international literature on patient safety [1,9,13,14].

On the one hand, research has found that hospitals with higher nurse burnout had worse patient outcomes, in terms of patient mortality, failure to rescue, and days of stay [16]. On the other hand, positive work environments attenuated the negative relationship between nurse burnout and patient outcomes [16]. Although the above study focused exclusively on nurses (typical in the extant safety research, with physicians as the other focus of attention), the findings are nonetheless poignant that investing in the positive work environment in hospitals could be beneficial for promoting staff well-being and protecting patient safety simultaneously. Another cross-sectional survey of cross-disciplinary health workers in a Taiwanese hospital group amid the COVID-19 pandemic found negative associations between teamwork climate, safety climate, and job satisfaction as aspects of the safety culture and staff burnout [4]. The available evidence tentatively suggests that safety culture may have a beneficial effect not only on patient outcomes, but also on employees’ well-being. Below, we will apply the tenet of the COR theory to propose one theoretically plausible account for the patient safety culture–staff well-being relationship: the institutional safety culture constituting a good work environment can act as a valuable work resource at multiple levels to protect staff well-being (reducing burnout and improving work–life balance) [2]. As patient safety culture encompasses multiple dimensions of the “good” work environment in terms of job satisfaction, supportive management, adequate work conditions, and strong teamwork, such resources should have ubiquitous well-being values for staff across professional disciplines and personal demography.

### 2.2. Patient Safety Culture as Work Resources: The Organizational Protectors for Staff Well-Being

Evolving from a general theory, the COR pivots on the central idea of resource. It posits that potential or actual loss of the valued resources is stressful for individuals [2]. This thus places the acquisition and facilitation of resources at a central position in the stress and adaptation process. Hobfoll broadly defined resources as those entities that either are centrally valued in their own right (e.g., self-esteem, close attachments) or are instrumental in obtaining centrally valued ends (e.g., money, social support) [2]. Furthermore, resource gains become more salient in the face of resource loss.

Medical work is very demanding and risk-ridden; thus, the effortful process of completing the daily work requires continuous resource investment by the individual [6,13]. Moreover, the clinical practice is inherently affected by a constellation of factors rooted at multiple levels of the medical institutions, including organizational/work environment (e.g., staffing and management support, safety climate), team (e.g., teamwork, task design, and collaboration), and personal characteristics (e.g., overconfidence) [17]. Existing research has firmly established that the demanding, challenging, and interdependent/cross-disciplinary medical work leads to resource depletion and stress which, in turn, increase the risk of professional burnout and diminished quality of life [1,6,13,15]. In this resource-depletion context, any mobilizable, congruent (those that “fit” the demands), and accessible resources are valuable to alleviate the negative effects of work in challenging conditions [18,19]. From the COR perspective, investments and interventions in patient safety can be viewed as valuable work resources operating at multiple levels of the medical organizations. Based on the theoretical analysis of risk factors in clinical practice [17], the improvement attempts on patient safety instill and foster resources (1) at the organizational level, such as building a prevailing safety climate, nurturing meaning of work and job satisfaction, strengthening management support, and improving working conditions; (2) at the team level, such as better designing and training for teamwork and collaboration; and (3) at the personal level, such as awareness for overconfidence and complacency under stress. These resources are represented in the six dimensions comprising the Safety Attitudes Questionnaire (SAQ): safety climate, job satisfaction, perception of management, working condition, teamwork climate, and stress recognition [20]. Together these multilevel resources could directly address the salient antecedents of burnout and diminished quality of life already identified in medical professions. For example, a systematic review on physician burnout has identified work overload, loss of support, limited collaboration, and unhelpful coping strategies as critical risk factors for burnout [6]. These can be alleviated to a certain extent through patient safety initiatives aiming at better management support, adequate working condition, strong teamwork, and realistic stress recognition. Similarly, a systematic review on nurse burnout has also noted that adverse job characteristics (e.g., high workload, low support, poor teamwork, negative work environment) are associated with burnout [5]. Furthermore, previous studies have found that hospital investment in the quality of care was associated with reduced burnout [3,4,21]; thus, emphasizing care quality and patient safety may act as work resources to protect staff well-being. We thus hypothesize:

**Hypothesis** **1** **(H1).***The patient safety culture is negatively related to staff burnout*.

**Hypothesis** **2** **(H2).***The patient safety culture is positively related to staff work–life balance*.

### 2.3. Ubiquity of the Value of Patient Safety Culture as Work Resources

The notion of safety culture is based largely on ideas about organizational culture, such that safety culture is defined as “those aspects of organizational culture which will impact on attitudes and behavior related to increasing or decreasing risk” [22]. Kayworth and Leidner have purported that organizational culture can be viewed as an organizational resource that facilitates knowledge management activities [23]. We further argue that safety culture as an important organizational resource can have universal value for all employees in the organization. Numerous organizational research has established the ubiquitous utility of work resources in ameliorating work strains and protecting employees’ well-being [18,19]. As interventions aiming at improving patient safety, reducing medical risks, and fostering a strong safety culture often take a high-level approach by investing in resources such as staffing, training, equipment, information systems, cross-disciplinary collaboration, communication, and coordination [1,4,24], these valuable resources should benefit ALL members of the organization. For example, in reviewing research on physician burnout, Patel et al. noted that work overload and insufficient interpersonal collaboration are two critical factors contributing to burnout [6]. Review of research on nurses also found that adverse job characteristics (e.g., high workload, low support, poor teamwork, negative work environment) are consistently associated with burnout [5]. Safety investments to reduce workload with better staffing, provide supportive management, and encourage cross-disciplinary collaborations are able to eradicate, to a certain extent, the abovementioned antecedents of staff burnout. Furthermore, in the context of the highly integrated healthcare organizations, safety culture as an institution resource should benefit diverse health professionals, not just physicians and nurses. We thus hypothesize:

**Hypothesis** **3** **(H3).***The negative association between patient safety culture and burnout is invariant for all staff (i.e., gender, age, managerial position, medical roles, seniority, direct patient contact, and incident reporting)*.

**Hypothesis** **4** **(H4).***The positive association between patient safety culture and work–life balance is invariant for all staff (i.e., gender, age, managerial position, medical roles, seniority, direct patient contact, and incident reporting)*.

## 3. Method

### 3.1. Procedure

This cross-sectional survey study was performed in the Taipei City Hospital (TPECH), approved by the Institutional Review Board (permit number: TCHIRB-11001021-E), and all subjects gave their consent to participate. TPECH has ten general and specialty hospitals located in the capital Taipei City of Taiwan. Being the largest healthcare group in Northern Taiwan, TPECH has 4.9 million patient contacts per year. Study participants were 5436 full-time staff working at TPECH at the point of study (October 2018). Data collection was conducted with institutional approval, using a web-based self-reported questionnaire during October–November 2018. Email invitations were sent to all hospital staff, with one follow-up reminder. Questionnaires were anonymous to ensure anonymity. At the end of the study, 3232 staff (response rate: 59.46%) completed the survey. No survey subject was excluded from the analysis. Although there were fewer responses from people over 61 years old (2.2%) compared to those from other age groups (under 30: 27.8%, 31~40: 32.3%, 41~50: 22.2%, 51~60: 15.5%), this is attributable to the low rate of older people working in medical institutions in Taiwan. Another check showed that there was no significant difference in the distribution of seniority in our sample. Thus, we conclude that although the survey was conducted on the web and the email was used for reminder, those methods did not result in the sampling bias of fewer cases of high-age or high-seniority people in the study.

Among the study sample, 2701 (83.6%) were female and 1941 (60.1%) were under the age of 40. Furthermore, 2572 (79.6%) had completed college education and 285 (8.8%) were managers. The sample profile of female majority and college graduates is typical of hospital staff in Taiwan [24]. Nearly all the respondents worked directly with patients (91.4%), but most (71.7%) had no safety incidents in the past year. The majority of respondents (63.3%) had tenure of less than 10 years. In term of medical division, 515 (15.9%) worked in critical care units, 1048 (32.4%) in inpatient wards, 673 (20.8%) in ambulatory clinics, and 996 (30.8%) in other hospital units. Overall, physicians accounted for 242 (7.5%) of the respondents, nursing staff accounted for 1718 (53.2%), and 1272 (39.4%) were other healthcare workers.

### 3.2. Measures

The questionnaire used in this study comprised 55 questions: 9 for demographic characteristics, 30 for patient safety culture, 9 for burnout, and 7 for work–life balance.

*Demographic Questions.* These questions comprised items on gender, age, job role, medical division, tenure at current institution, education, managerial position, frequency of direct patient contact, and reporting of any safety incidents in the past 12 months. For all further analyses, job roles were grouped into doctors, nurses, and other hospital staff (e.g., allied health professionals, pharmacists, administrative and support workers). Medical divisions were classified as critical care units (e.g., ICU, operation rooms, and emergency room), inpatient wards, ambulatory clinics, and others (e.g., administration, auxiliary, and support units).

*Patient Safety Culture.* The Safety Attitudes Questionnaire (SAQ) is the most commonly used safety culture measure in the literature [20]. The Chinese version has been adopted and validated by the Joint Commission of Taiwan (JCT) for the Taiwanese medical setting and used to establish a nationwide profile on patient safety in all ranges of hospitals since 2009, including the study institution (TPECH) [24,25]. The SAQ comprises 30 questions across six dimensions: 6 for teamwork climate (e.g., I have the support I need from other personnel to care for patients), 7 for safety climate (e.g., I am encouraged by my colleagues to report any patient safety concerns I may have), 5 for job satisfaction (e.g., This hospital is a good place to work), 4 for stress recognition (e.g., I am more likely to make errors in tense or hostile situations), 4 for perception of management (e.g., The levels of staff in this clinical area are sufficient to handle the number of patients), and 4 for working conditions (e.g., The hospital does a good job of training new personnel). Subjects responded on a 5-point Likert scale, ranging from 1 (strongly disagree) to 5 (strongly agree). Following the common practice in patient safety literature, all SAQ items were summed with higher scores representing more positive attitude towards patient safety culture [20]. Cronbach’s α for the SAQ was 0.90 in this study.

*Burnout.* The Maslach burnout inventory (MBI) is the most commonly used burnout measure for healthcare workers in the literature [26]. According to past research, emotional exhaustion is the core component of burnout compared with other dimensions (depersonalization and personal accomplishment), and the most obvious manifestation of the syndrome [27]. We used the 9-item emotional exhaustion subscale from MBI to index burnout in the present study. The Chinese version has been adopted and validated by the JCT for the Taiwanese medical setting, to be incorporated with SAQ in nationwide surveys of hospitals [4,28]. Exemplary items are “Working with people all day is really a strain for me” and “I feel burned out from my work”. Items are scored on 5-point Likert scales ranging from strongly disagree (1) to strongly agree (5). Higher scores indicate higher levels of burnout. In the present study, the internal consistency reliability was 0.94.

*Work–Life Balance (WLB)*. The work–life climate scale was a behavior-based measure specifically developed by Sexton et al. for healthcare workers [9]. It evaluates behavioral frequencies of work–life (im)balance such as skipping meals, not taking breaks, and changing personal plans because of work. This measure has shown strong psychometric properties and good alignment with organizational culture constructs such as the safety culture [9,28,29]. The Chinese version has been adopted and validated by the JCT for the Taiwanese medical setting, to be incorporated with SAQ in nationwide surveys of hospitals [28]. Exemplary items are “Worked through a day/shift without any breaks” and “Slept < 5 h in a night”. Participants reported behavioral frequencies as: “1” for rarely or none of the time (less than 1 day); “2” for some or a little of the time (1–2 days); “3” for occasionally or a moderate amount of time (3–4 days); and “4” for all of the time (5–7 days). Cronbach’s α in this study was 0.90.

### 3.3. Strategy of Analysis

All the statistical analyses were performed using the software IBM SPSS 25 (New York, NY, USA) and AMOS 24 (New York, NY, USA). We adopted the two-step approach, evaluating the measurement models followed by the structural model evaluation [30]. That is, we first conducted a confirmatory factor analysis (CFA) to verify the factor structure by confirming that each measure is loaded on a particular factor [31]. We also checked for the common method variance bias, as our data are all self-reported [32]. We then tested hypotheses 1 and 2 through structural equation modeling (SEM) based on the maximum likelihood estimation method. Hypotheses 3 and 4 (i.e., the equivalence of model across groups) were tested through multi-group structural equation modeling analyses [33].

## 4. Results

### 4.1. Descriptive Analysis

Prior to the hypotheses testing, bivariable correlations were computed among all the research variables, and results are shown in Table 1. Male gender significantly positively correlated with overall patient safety culture and work–life balance, but negatively correlated with burnout. Older age positively correlated with overall patient safety culture, but negatively correlated with burnout and work–life balance. Being a manager positively correlated with patient safety culture, but negatively correlated with work–life balance. Having higher education, direct patient contact, and reported incidents significantly negatively correlated with patient safety culture and work–life balance, while positively correlated with burnout. Longer tenure significantly positively correlated with burnout, but negatively correlated with work–life balance. Finally, all six SAQ dimensions of patient safety culture significantly positively correlated with one another.

### 4.2. Hypothesis Testing

We conducted confirmatory factor analysis (CFA) to test for convergent and divergent validity on eight latent constructs (six first-order patient safety culture dimensions, burnout, work–life balance) with 46 observed indicators. Results revealed that all first-order scale items loaded significantly (*p* < 0.001, all factor loadings > 0.50) on their designated constructs, indicating acceptable individual item reliability. The composite reliability (CR) for the eight constructs ranged from 0.90 to 0.96, indicating good internal consistency of all constructs. Finally, the average variances extracted (AVE) for the eight constructs ranged from 0.59 to 0.82, indicating acceptable convergence of observed indicators to their designated constructs. According to Hair et al. [34], all eight constructs in our study demonstrated good convergent validity.

To justify the collapsing of all the SAQ dimensions into one overall score, we tested the measurement quality of the second-order construct of patient safety culture (model 1 in Table 2). The CFA results of six-factor model (second-order) showed that all the fit indexes were within acceptable ranges, except for the χ^2^/df value (χ^2^ = 5194.52; df = 399; χ^2^/df = 13.02; comparative fit index (CFI) = 0.95; and root-mean-square error of approximation (RMSEA) = 0.06). Given the large sample size of the current study, the model fit was considered satisfactory [34]. For the sake of parsimony, we thus merged six dimensions of SAQ into one higher-level concept of patient safety culture in all further analyses.

In order to test for discriminant validity and to rule out common method variance for self-report data, we compared our hypothesized three-factor model (measurement model: second-order patient safety culture, burnout, work–life balance) against three alternative models: two-factor model (combining burnout and work–life balance), one-factor model (combining patient safety culture, burnout, and work–life balance), and null model (no latent variable). The summary of model comparisons is presented in Table 2 (models 2 to 5). The results showed that the measurement model (χ^2^ = 6428.80, df = 206, χ^2^/df = 31.21, CFI = 0.90, RMSEA = 0.10) fit the data best, and outperformed any simpler representations of the data (*p* < 0.01 for all model comparisons). We thus ensured that all variables were distinct constructs, and the results were not likely caused by common method variance.

The results of structural equation model analysis for testing hypotheses 1 and 2 are shown in Table 3. It can be seen that the nonstandardized coefficient of patient safety culture on burnout was −0.91 (*p* < 0.001), indicating that patient safety culture had a significant negative impact on burnout: with higher patient safety culture, staff burnout becomes lower. Therefore, hypothesis 1 was supported. The nonstandardized coefficient of patient safety culture on work–life balance is 0.40 (*p* < 0.001), indicating that patient safety culture was positively related to staff work–life balance. Hence, hypothesis 2 was also supported.

To test for the invariance of the hypothesized structural relationships between safety culture and burnout and work–life balance, across staff groups of different gender, age, managerial position, medical roles, seniority, having/not having direct patient contact, or incident reporting (H3 and H4), multi-group analyses were conducted. A model (structural weights model) that restricted the paths to be the same between the two groups (e.g., male and female, or nurse and non-nurse) and a model (unconstrained model) with no restrictions were compared [35]. We can obtain a difference of chi-square value by subtracting the chi-square value of the unconstrained model from the chi-square value of the structural weights model, and its degree of freedom is the difference between the degrees of freedom of the two models. If the difference in chi-square value is not significant, then it can be inferred that the model has invariance. If the difference in chi-square value is significant, we cannot accept the assumption that the path coefficients are equal across different groupings. Since the chi-square value is easily affected by the size of the sample, indicators such as ΔCFI and ΔTLI (Tucker–Lewis index) need to be checked to increase the accuracy of the judgment. When ΔCFI ≦ 0.01 or ΔTLI ≦ 0.02, it means that the model has multi-group invariance [36]. A summary of this series of analyses is presented in Table 4.

The results in Table 4 show that path I of this study (patient safety culture → burnout) has statistically significant differences in the chi-square values for gender, incident reporting, job role, tenure, and patient contact; path II (patient safety culture → work–life balance) has statistically significant differences in the chi-square values for age, incident reporting, and patient contact. Both sets of results seem to suggest that there may be some differences in the paths between the different groups. However, given our large sample size, we turned to the alternative fit indices for more informative diagnosis. Since both ΔCFI and ΔTLI reached the recommended values (ΔCFI ≦ 0.01 and ΔTLI ≦ 0.02), we concluded that the hypothesized structural model had multi-group invariance, namely, its paths were largely the same for different groups of staff, whether categorized in demographical or job characteristics.

## 5. Discussion

The present study tested COR theory-based hypotheses that patient safety culture has a positive relationship with staff well-being across all healthcare disciplines. We indeed found that patient safety culture had a negative relationship with staff burnout, but a positive one with staff work–life balance. More importantly, these relationships were the same for various groups of staff, regardless of their personal and job factors. In other words, we found full support for all of our hypotheses.

### 5.1. Theoretical Implications

Our findings have two important theoretical implications. First, we introduce the conservation of resources theory to the safety literature in explaining how institutional patient safety culture leads to better staff well-being by reducing burnout and enhancing work–life balance. Our findings complement previous studies examining the relationship between nurse burnout and patient safety, which framed staff burnout as an antecedent of perceived safety [13,14]. Conceptualizing patient safety culture as an institution-wide resource operating across the organization at team and individual level, we offer a distinct view on hospital quality intervention as a pivotal mechanism to meet the dual challenge of improving patient safety and staff well-being in healthcare institutions worldwide. Thus, we not only contribute to the scarcity of empirical studies on the association between patient safety and staff well-being, but also address the theoretical void in the extant literature on patient safety and professional burnout [1]. Future research may build on the conservation of resources model to delineate the exact paths leading from safety culture to staff well-being. For instance, when resources are abundant, what resource investment strategies do staff use to reduce workload and facilitate collaboration, in order to reduce burnout and improve work–life balance? A recent study of US healthcare workers within a large academic healthcare system (N = 10,627) found that positive work–life climate was associated with better teamwork and safety climates, as well as lower personal burnout and burnout climate [29]. Thus, work–life balance as a positive indicator for well-being deserves more research and interventional attention.

Second, we contribute to the burnout literature by extending our findings to all healthcare disciplines. The extant literature on both patient safety and health workers’ well-being has predominantly focused on professional burnout, especially those of physicians and nurses [1,5,6]. Including work–life balance as a positive indicator for well-being, the present study fills the gap on the paucity of studies examining multiple outcomes of the quality of life for health workers [9]. More importantly, we verify the ubiquity of the structural relationship between safety culture and staff well-being for all healthcare workers, despite their personal and job characteristics. This provides an evidence base for investing organizational resources on improving quality of care to benefit all employees across all disciplines [3]. Our findings are especially relevant in the post-pandemic restructuring and restrategizing of healthcare institutions. We demonstrate that the culture of patient safety can provide a solid foundation for both hospital and staff resilience.

### 5.2. Managerial Implications

We confirmed that institutional patient safety culture is beneficial for reducing burnout and enhancing work–life balance for all hospital staff, irrespective of individual and job characteristics. Therefore, investment in patient safety is “a good business” for medical institutions. These initiatives as work resources can be planned at the organizational, team, and individual level to reduce staff burnout and increase quality of life. Things that can be actioned include increasing staffing and equipment, providing on-the-job training, communicating with managers without barriers, encouraging and subsidizing employees’ leisure activities after work, and providing psychological counseling. In terms of work–life balance, hospitals can provide childcare, hosting family days, especially for the spouses of nonmedical employees, to help them understand employees’ hard work and thus increase family support.

### 5.3. Limitation and Future Research Directions

This study has several limitations that may affect the generalizability of our findings. First, our study is a cross-sectional design; thus, no causal inferences should be made pertaining to the relationships among study variables. Although our postulation of patient safety culture as a protective organizational resource pivots on the well-tested arguments of the COR theory and its rich literature, we are unable to empirically tease out the prospective effects of patient safety on staff well-being. While longitudinal studies are very rare in patient safety research, researchers should endeavor to establish the casual order of patient safety and staff well-being in the future. As suggested in one prospective study, the relationship may be reciprocal when the temporal effect between safety perception and personal well-being is modeled [15]. A second limitation is that we conducted the survey research in only one hospital system, the Taipei City Hospital (TPECH). There are other large-scale medical centers and regional hospitals with different characteristics in the Taipei metropolitan area, e.g., central-government-affiliated or profit-oriented hospitals. The governance and work environment are apparently not similar among these organizations; we, therefore, would not claim that our findings are generalizable to the diverse hospitals in Taiwan, not to mention other countries or regions in the world. It is not the purpose of our study to make direct comparisons of the levels of patient safety culture and/or staff well-being across personnel groups, organizations, or countries; we instead focused on the essence of the relationship between patient safety culture and staff well-being. The nature of the structural relationship between two constructs is independent of their magnitudes on the measurement scales. We found preliminary evidence that the negative relationship between patient safety culture and staff burnout, as well as the positive relationship between patient safety culture and staff work–life balance, is uniform for a large cross-disciplinary staff within one hospital system. We acknowledge that there were differences in patient safety and well-being across different job roles in our survey data. Specifically, doctors reported significantly more positive safety attitudes than nurses and other healthcare workers, while nurses reported the highest exhaustion, and those in other healthcare roles had the best work–life balance. Due to their functions and job tasks, each personnel group contributes differently in building a safety culture and it is worth more nuanced examination into the relationship between this organizational resource and performance/well-being for each group. Future research could extend and test our findings in diverse work settings of different healthcare organizations, or even different countries. Finally, while it is valuable to include both negative (burnout) and positive (work–life balance) indicators of staff well-being in the context of patient safety, as we did, future studies should include objective measurements of staff performance and patient outcomes (e.g., mortality, failure to rescue, and hospital stay duration). Validated against specific patient safety outcome (e.g., anesthesia-related mortality), we can obtain more comprehensive knowledge regarding whether we are dealing with different safety issues on the same basis of staff well-being. Therefore, hospitals can target resources more precisely to maximize the invention effects in quality improvement.

## 6. Conclusions

To conclude, the present study is among few exploring the link between patient safety culture and staff well-being and is the first conducted in Taiwan. To our knowledge, it is also the first to conceptualize patient safety culture as an organizational resource for hospital staff of all disciplines. Our results suggest that patient safety culture is a valuable work resource protecting staff well-being by reducing burnout and increasing work–life balance. Furthermore, such protective value is universal for all healthcare disciplines and staff of diverse demography and job characteristics. These findings support the argument that nurturing the institutional safety culture is fostering the critical resources for improving staff well-being, thus contributing to building and sustaining a healthy and resilient workforce. Healthcare managers can consider a simpler measure of this resource in ensuring adequate staffing levels across all departments of the institution [1]. In the post-pandemic era, healthcare institutions should restrategize to cultivate a supportive and safe environment for the welfare of both patients and the staff. For example, managers should divert more resources to help nurses and those of ethnic minorities to cope with stress and burnout caused by the pandemic [4,37]. Now is the time to turn the crisis of COVID-19 into an opportunity for building hospital and individual resilience.

## Figures and Tables

**Table 1 ijerph-19-03722-t001:** Correlation matrix (including mean, SD, α).

	Mean	SD	1	2	3	4	5	6	7	8	9	10	11	12	13	14	15	16
1. Gender																		
2. Age			0.04 *															
3. Managerial Position			0.13 **	0.23 **														
4. Incident Reporting			−0.04 *	−0.10 **	0.12 **													
5. Tenure			−0.03	0.53 **	0.24 **	0.02												
6. Education Years			0.05 **	−0.21 **	0.17 **	0.17 **	0.01											
7. Patient Contact			−0.03	−0.05 **	0.01	0.14 **	0.01	0.11 **										
8. Teamwork Climate	4.11	0.75	0.08 **	0.13 **	0.15 **	−0.07 **	0.03	−0.05 **	−0.09 **	(00.79)								
9. Safety Climate	4.06	0.77	0.09 **	0.13 **	0.15 **	−0.06 **	0.02	−0.04 *	−0.09 **	0.90 **	(00.82)							
10. Job Satisfaction	3.93	0.91	0.10 **	0.16 **	0.14 **	−0.12 **	0.03	−0.08 **	−0.09 **	0.81 **	0.83 **	(00.90)						
11. Stress Recognition	3.87	0.92	0.11 **	−0.05 **	0.07 **	0.08 **	−0.01	0.07 **	0.07 **	0.21 **	0.23 **	0.18 **	(00.83)					
12. Perception of Management	3.99	0.85	0.08 **	0.11 **	0.13 **	−0.07 **	0.01	−0.06 **	−0.09 **	0.83 **	0.86 **	0.82 **	0.22 **	(00.87)				
13. Working Condition	3.86	0.85	0.08 **	0.12 **	0.12 **	−0.07 **	0.04 *	−0.04 *	−0.11 **	0.79 **	0.82 **	0.80 **	0.23 **	0.83 **	(00.84)			
14. Patient Safety Culture (Overall)	3.97	0.69	0.11 **	0.12 **	0.15 **	−0.06 **	0.02	−0.04 *	−0.08 **	0.91 **	0.93 **	0.89 **	0.44 **	0.91 **	0.90 **	(00.87)		
15. Burnout	2.48	0.95	−0.09 **	−0.05 **	−0.02	0.14 **	0.06 **	0.08 **	0.12 **	−0.63 **	−0.65 **	−0.70 **	−0.13 **	−0.67 **	−0.71 **	−0.70 **	(00.81)	
16. Work–Life Balance	3.13	0.66	0.10 **	−0.05 **	−0.07 **	−0.15 **	−0.12 **	−0.08 **	−0.17 **	0.35 **	0.36 **	0.38 **	0.09 **	0.39 **	0.42 **	0.40 **	−0.52 **	(0.77)

Notes: Square root of AVE appears in brackets on the diagonal. Gender (female = 0, male = 1); age (≤40 = 0, >40 = 1); managerial position (no = 0, yes = 1); incident reporting (none = 0, at least one = 1); tenure (≤10 years = 0, >10 years = 1); patient contact (no = 0, yes = 1). * *p* < 0.05, ** *p* < 0.01.

**Table 2 ijerph-19-03722-t002:** Measurement model: summary of model comparisons.

	Model	χ²	df	χ²/df	Δχ²	Δdf	CFI	TLI	RMSEA	SRMR	AIC
1. Six-factor Model (Second-Order SAQ)	Patient safety culture and its six dimensions	5194.52	399	13.02	391.50 ***	9	0.95	0.95	0.06	0.03	173,635.49
2. Three-factor Model (Hypothesized Model)	Patient safety culture, burnout, and work–life balance	6428.80	206	31.21	-	-	0.90	0.89	0.10	0.05	6522.80
3. Two-factor Model	Patient safety culture, and the combination of burnout and work–life balance	14,823.73	208	71.27	8394.92 ***	2	0.77	0.74	0.15	0.11	14,913.73
4. One-factor Model	The combination of patient safety culture, burnout and work–life balance	23,498.22	209	112.43	17,069.42 ***	3	0.63	0.59	0.19	0.13	23,586.22
5. Null Model	No latent variable	63,441.58	231	274.64	57,012.77 ***	25	0.00	0.00	0.29	0.47	63,485.58

Notes: SAQ = Safety Attitudes Questionnaire. *** *p* < 0.001.

**Table 3 ijerph-19-03722-t003:** Path analysis of the research model.

	Burnout			Work–Life Balance		
	B	SE	*β*	B	SE	*β*
Patient Safety Culture	−0.91 ***	0.02	−0.74	0.40 ***	0.02	0.44
R^2^	0.55			0.19		

Note: *** *p* < 0.001.

**Table 4 ijerph-19-03722-t004:** Summary of the multi-group analysis.

	Path I Patient Safety Culture → Burnout				Path II Patient Safety Culture → Work–Life Balance			
	Δχ^2^	Δdf	ΔCFI	ΔTLI	Δχ^2^	Δdf	ΔCFI	ΔTLI
Gender	17.59 ***	1	<0.001	<0.001	1.21	1	<0.001	<0.001
Age	0.14	1	<0.001	<0.001	9.70 **	1	<0.001	<0.001
Managerial Position	2.91	1	<0.001	<0.001	3.70	1	<0.001	<0.001
Incident Reporting	4.96 *	1	<0.001	<0.001	5.94 *	1	<0.001	<0.001
Job Role	26.46 ***	1	<0.001	<0.001	3.74	1	<0.001	<0.001
Tenure	4.64 *	1	<0.001	<0.001	0.01	1	<0.001	<0.001
Patient Contact	54.19 ***	1	<0.001	<0.001	11.60 **	1	<0.001	<0.001

Notes: Gender (female = 0, male = 1); age (≤40 = 0, >40 = 1); managerial position (no = 0, yes = 1); incident reporting (none = 0, at least one = 1); job role (nurse = 0, non-nurse = 1); tenure (≤10 years = 0, >10 years = 1); patient contact (no = 0, yes = 1). * *p* < 0.05, ** *p* < 0.01, *** *p* < 0.001.

## Data Availability

The data that support the findings of this study are available upon reasonable request to the Taipei City Hospital or the corresponding author.

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
