# Peer review of "Patient Safety and Staff Well-Being: Organizational Culture as a Resource"

_ijerph, 2022, doi:10.3390/ijerph19063722_

Round 1
Reviewer 1 Report
Congratulations on the study. Although at first it may seem like an obvious and unoriginal study, the authors present an innovative approach. They show the knowledge that will be added to the topic after this scientific investigation. The limitations of the study are also pointed out. Despite having been carried out before the COVID-19 pandemic, they discuss the relevance and applicability of these findings in the current period.
Author Response
Thank you for the encouraging feedback.
Reviewer 2 Report
Title: Patient safety and staff well-being
Authors: Lu et al
MS No: ijerph-1630831
In this interesting and well-written manuscript, the authors described a relationship between patient safety culture and healthcare workers’ well-being (an extension research from the reference 4). Using a study methodology and analysis, the authors found such patient safety culture was negatively related to HCW’s burnout, in contrast to staff work-life balance. Such relationship is independent of staff demography and job characteristics. It is concluded that patient safety culture is vital and crucial to the hospital and institute, and equally to a large cross-disciplinary staff.
Comments:
Introduction section:
- Is there any difference between oriental and western people (in particular, hospital staff) regarding the concept of “well-being”, the threshold of “work burn-out”, the degree of “work-life balance”? (e.g., references 1, 11,12)
- In Taiwan (also in Taipei metropolitan), the hospitals run their clinical service under the unique umbrella of “universal health coverage”. Under this unique condition and weighted by the GDP in each country, is it possible that the definition of the hospital staff “well-being” in Taiwan be different from those in other countries? For example, nurses and medical residents in Taipei (or in Taiwan) suffered more (or less) work burn-out than those in other countries?
- Regarding the patient safety outcome (e.g., mortality, failure to rescue and hospital stay duration), how different such patient safety outcome in Taiwan is from other countries? Or is it different among each region in Taiwan (metropolitan area versus rural area)? For example, it has been documented that the anesthesia-related mortality is 17 times higher than those in Japan (data from Taiwan Society of Anesthesiology). Therefore, are we dealing with different patient safety outcome on the same basis of staff well-being?
- The authors conducted a survey research in Taipei City Hospitals and expected such results obtained from this study can be generalized to those in other hospitals and medical centers in Taipei metropolitan, Taiwan, and possibly other countries or regions in the world. Meanwhile, there are other large-scale medical centers hospitals with different characteristics in Taipei metropolitan areas, e.g., central government affiliated or profit-oriented hospitals. The CMI and workload apparently are not similar among each other in Taipei metropolitan areas. Therefore, is such generalizability obtained from this study expected to sustain for those hospitals in comparison to the city government-run hospitals system ?
Methods:
- (a.3.2.) Measures: About “burnout” and 9-item emotional exhaustion subscale from MBI (Maslach burnout inventory), is it possible to describe a little more about this measurement? Is this scale validated for Taiwanese or Chinese people?
- (b.3.2.) Measures: Similarly, about WLB (Work-Life Balance, cited from “Burnout in the NICU setting and its relation to safety culture. BMJ Quality & Safety.2014;23(10):806-813.”), is this 7-item survey scale validated for Chinese or Taiwanese users?
Discussion:
- In the last paragraph, the authors concluded that "Our findings are especially relevant in the post-pandemic restructuring and re-strategizing of healthcare institutions." "In the post-pandemic era, healthcare institutions should re-strategize to cultivate …………" It is not clear how the results from this survey research be related to the COVID pandemic incident. Namely, before and after the COVID pandemic, should we expect any different strategies for patient safety culture and staff well-being?

Author Response
Revisions made following Reviewer 2’s suggestions
Comments:
Introduction section:
- Is there any difference between oriental and western people (in particular, hospital staff) regarding the concept of “well-being”, the threshold of “work burn-out”, the degree of “work-life balance”? (e.g., references 1, 11,12)
Response: There is extensive research on the cultural connotation and conceptualization of “well-being” in the positive/social psychology literature. In the present study, however, we adopt the same conceptualization and measurements of staff well-being, burnout, and work-life balance, as commonly used in the international literature on patient safety. This decision is made to converse and contribute to the mainstream research on healthcare quality. I have made it explicit in this revision. (p.2)
- In Taiwan (also in Taipei metropolitan), the hospitals run their clinical service under the unique umbrella of “universal health coverage”. Under this unique condition and weighted by the GDP in each country, is it possible that the definition of the hospital staff “well-being” in Taiwan be different from those in other countries? For example, nurses and medical residents in Taipei (or in Taiwan) suffered more (or less) work burn-out than those in other countries?
Response: Taiwan does have a unique national health insurance system (NHIS) that is accessible to every citizen. I have now cited new references to state that despite the different governance and conditions for clinical practices, available cross-country data seem to suggest that Taiwanese physicians and nurses suffered no more (or less) burnout than those in other countries (Chen, 2013; Woo et al., 2020). (p.2) For example, Chen et al. (2013) found that 36% of Taiwanese doctors suffered from burnout, compared to the prevalence rate of 30%~46% for US physicians.
- Regarding the patient safety outcome (e.g., mortality, failure to rescue and hospital stay duration), how different such patient safety outcome in Taiwan is from other countries? Or is it different among each region in Taiwan (metropolitan area versus rural area)? For example, it has been documented that the anesthesia-related mortality is 17 times higher than those in Japan (data from Taiwan Society of Anesthesiology). Therefore, are we dealing with different patient safety outcome on the same basis of staff well-being?
Response: Our survey data are not linked to hospital performance per patient outcomes. I have now acknowledged this as a limitation and have urged future studies to validate our findings against specific patient safety outcome (e.g., anesthesia-related mortality). Doing so can generate more comprehensive knowledge as to whether we are dealing with different safety issues on the same basis of staff well-being. (p.10, last point in the new added “5.3. Limitation and Future Research Directions” section)
- The authors conducted a survey research in Taipei City Hospitals and expected such results obtained from this study can be generalized to those in other hospitals and medical centers in Taipei metropolitan, Taiwan, and possibly other countries or regions in the world. Meanwhile, there are other large-scale medical centers hospitals with different characteristics in Taipei metropolitan areas, e.g., central government affiliated or profit-oriented hospitals. The CMI and workload apparently are not similar among each other in Taipei metropolitan areas. Therefore, is such generalizability obtained from this study expected to sustain for those hospitals in comparison to the city government-run hospitals system ?
Response: In the new added “5.3. Limitation and Future Research Directions” section, I have acknowledged this as a limitation (second point, p.10). I have also made it explicit that we do not expect that our findings are generalizable to the diverse hospitals in Taiwan, not to mention other countries or regions in the world.
Methods:
- (a.3.2.) Measures: About “burnout” and 9-item emotional exhaustion subscale from MBI (Maslach burnout inventory), is it possible to describe a little more about this measurement? Is this scale validated for Taiwanese or Chinese people?
Response: Yes, the Chinese version has been validated for the Taiwanese medical setting, and is in wide use. I have now added those references along with more description of the scale. (p.5-6)
- (b.3.2.) Measures: Similarly, about WLB (Work-Life Balance, cited from “Burnout in the NICU setting and its relation to safety culture. BMJ Quality & Safety.2014;23(10):806-813.”), is this 7-item survey scale validated for Chinese or Taiwanese users?
Response: Yes, the Chinese version has been validated for the Taiwanese medical setting. I have now added the supporting reference. (p.6)
Discussion:
- In the last paragraph, the authors concluded that "Our findings are especially relevant in the post-pandemic restructuring and re-strategizing of healthcare institutions." "In the post-pandemic era, healthcare institutions should re-strategize to cultivate …………" It is not clear how the results from this survey research be related to the COVID pandemic incident. Namely, before and after the COVID pandemic, should we expect any different strategies for patient safety culture and staff well-being?
Response: I have now elaborated on the statement with emerging evidence from recent studies after the pandemic. (p.11)

Reviewer 3 Report
This study finding has two important theoretical implications.
First, it introduce the conservation of resources theory to the safety literature in explaining how institutional patient safety culture leads to better staff well-being by reducing burnout and enhancing work-life balance. The findings complement previous studies examining the relationship between nurse burnout and patient safety, which framed staff burnout as an antecedent of perceived safety [11,12].
This study applied the resources theory to patient safety and found that better staff well-being by reducing burnout and enhancing work-life balance was related to patient safety culture in the healthcare institute. The research topic is interesting and the study results are expected.
-
The abstract of this study does not present clear findings, only brief descriptions, and does not show the relevant important statistical results.
-
The keywords below the abstract are in uppercase or lowercase, please modify according to the requirements of the journal.
-
This study used web-based self‐reported questionnaire during October to November in 2018. Total participants were all 5,436 full-time staff working at TPECH at the point of study (October 2018). Using E-mail invitations were sent by to all hospital staff, with one follow-up reminder. Questionnaires were anonymous to ensure anonymity. At the end of the study, 3,232 staff (response rate: 59.46%) completed the survey.
Is there any survey subject excluded in this study, because of the use of mail, whether the main recovered object has the problem of fewer cases of high age or high seniority people? Or an explanation to strengthen the research limitation.
-
The IRB approved by the Institutional Review Board, Taipei City Hospital Institutional Review Board (Permit number: TCHIRB-11001021-E) had shown in the end of this paper “Institutional Review Board Statement”, but it still should write at the method part (Page 6, 3.1. Procedure part first paragraph).
The “Ethics approval and consent to participate” should be a simple description in the manuscript, about IRB (Institutional Review Board) / IEC (Independent Ethics Committee) approve or the data is applied for or authorized to use.
-
There is innovative research design and research method application in this study. It is a cross-sectional study; the limitation part should be included in the manuscript.
-
In the Method and Result part, please provide the full text of all the abbreviations used in the manuscript text and figure legends/tables. All abbreviations should still be defined in the text at first use.. Please check all the text of the manuscript.
- The work-life balance is a positive indicator for well-being, and it looks like the important variable for patient safety in the hospital, the author mentioned the finding of this study reinforced the few exploring the link between patient safety culture and staff well-being. Based on this argument, if it is possible, please write more clearly in the discussion part, also the authors should try to compare with more other related newer references.
-
Some of the research content may be similar to the following literature, and it is recommended to review the full text to avoid excessive self-citation or the risk of plagiarism.
Chen, H. Y., Lu, L., Ko, Y. M., Chueh, J. W., Hsiao, S. Y., Wang, P. C., & Cooper, C. L. (2021). Post-Pandemic Patient Safety Culture: A Case from a Large Metropolitan Hospital Group in Taiwan. International Journal of Environmental Research and Public Health, 18(9), 4537.

Author Response
Revisions made following Reviewer 3’s suggestions
- The abstract of this study does not present clear findings, only brief descriptions, and does not show the relevant important statistical results.
Response: I have now elaborated on the findings adding statistical results. (p.1)
- The keywords below the abstract are in uppercase or lowercase, please modify according to the requirements of the journal.
Response: I have now revised the keywords all in lowercase.
- Is there any survey subject excluded in this study, because of the use of mail, whether the main recovered object has the problem of fewer cases of high age or high seniority people? Or an explanation to strengthen the research limitation.
Response: No survey subject was excluded from the analysis. I have now added some description of the sample profile per age/seniority, to explain that the fewer responses from the older group was due to their low representation in the practice, rather than a methodological flaw. (p.5).
- The IRB approved by the Institutional Review Board, Taipei City Hospital Institutional Review Board (Permit number: TCHIRB-11001021-E) had shown in the end of this paper “Institutional Review Board Statement”, but it still should write at the method part (Page 6, 3.1. Procedure part first paragraph).
Response: The statement has been added in the main text. (p.4)
- The “Ethics approval and consent to participate” should be a simple description in the manuscript, about IRB (Institutional Review Board) / IEC (Independent Ethics Committee) approve or the data is applied for or authorized to use.
Response: The statement has been added in the main text. (p.4)
- There is innovative research design and research method application in this study. It is a cross-sectional study; the limitation part should be included in the manuscript.
Response: In the new added “5.3. Limitation and Future Research Directions” section, I have acknowledged this as a limitation (first point, p.10).
- In the Method and Result part, please provide the full text of all the abbreviations used in the manuscript text and figure legends/tables. All abbreviations should still be defined in the text at first use. Please check all the text of the manuscript.
Response: The full text of all the abbreviations have now been added in the text at first use. (p.7, 8)
- The work-life balance is a positive indicator for well-being, and it looks like the important variable for patient safety in the hospital, the author mentioned the finding of this study reinforced the few exploring the link between patient safety culture and staff well-being. Based on this argument, if it is possible, please write more clearly in the discussion part, also the authors should try to compare with more other related newer references.
Response: The point is taken up in the discussion, and our results are now compared those from a recent study (Schwartz et al., 2019). (p.9)
- Some of the research content may be similar to the following literature, and it is recommended to review the full text to avoid excessive self-citation or the risk of plagiarism.
Chen, H. Y., Lu, L., Ko, Y. M., Chueh, J. W., Hsiao, S. Y., Wang, P. C., & Cooper, C. L. (2021). Post-Pandemic Patient Safety Culture: A Case from a Large Metropolitan Hospital Group in Taiwan. International Journal of Environmental Research and Public Health, 18(9), 4537.
Response: Thank you for the reminder. I have now reviewed the full text to re-phrase/re-write, avoiding excessive overlaps with the published contents.

Reviewer 4 Report
Dear Authors,
I would like to congratulate you for proving the concept that in the operation of a medical entity there must be a culture of ensuring patient safety, treated as a specific, important resource of this institution. Undoubtedly, it is worth assessing and publishing the idea that organizational and sociological intangible assets of medical entities (particularly, at hospitals) must be treated and developed as another resource, after material values. This thesis is confirmed by the research carried out with the use of a standardized questionnaire. However, I would like to highlight the importance of this resource for different medical professionals. Research hypotheses assessed with grouped results confirm your assumptions, however, medical facilities employ different medical professionals (doctors and nurses are already two groups) who have clearly different tasks. The results shown for the effects of patient safety culture on individual groups in terms of burnout and work-life balance may (most likely) differ for each group of respondents. I suggest checking the survey results for each of the three respondent groups (physicians, nurses, others). Effort and commitment of each group in building a safety culture resource and the impact of this resource on the personnel is different for each group (due to their functions and tasks).
Moreover, the conclusions lack practical guidance for healthcare managers on what actions should be taken to increase the safety culture resource. Are there simpler measures of this resource without the need to conduct such a large survey with a questionnaire?
Author Response
Revisions made following Reviewer 4’s suggestions
- …However, I would like to highlight the importance of this resource for different medical professionals. Research hypotheses assessed with grouped results confirm your assumptions, however, medical facilities employ different medical professionals (doctors and nurses are already two groups) who have clearly different tasks. The results shown for the effects of patient safety culture on individual groups in terms of burnout and work-life balance may (most likely) differ for each group of respondents. I suggest checking the survey results for each of the three respondent groups (physicians, nurses, others). Effort and commitment of each group in building a safety culture resource and the impact of this resource on the personnel is different for each group (due to their functions and tasks).
Response: The point is taken up in the discussion, and I have elaborated on the implications of patient safety for different job groups. I have also acknowledged that there were differences on patient safety and well-being across different job roles in our survey data. Specifically, doctors reported significantly more positive safety attitudes than nurses and other healthcare workers, while nurses reported the highest exhaustion and those in other healthcare roles had the best work-life balance. (p.10, before the last point in the new added “5.3. Limitation and Future Research Directions” section)
- Moreover, the conclusions lack practical guidance for healthcare managers on what actions should be taken to increase the safety culture resource. Are there simpler measures of this resource without the need to conduct such a large survey with a questionnaire?
Response: I have now added some suggestion per your point, as an exemplar. (p.11)
